# Biogeophysical Effects of Land-Use and Land-Cover Changes in South Asia: An Analysis of CMIP6 Models

**Juliana Freitas Santos \*, Udo Schickhoff** 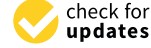**, Shabeh ul Hasson** **and Jürgen Böhner**

Center for Earth System Research and Sustainability (CEN), Institute of Geography at University of Hamburg, Bundesstraße 55, 20146 Hamburg, Germany; udo.schickhoff@uni-hamburg.de (U.S.); shabeh.hasson@uni-hamburg.de (S.u.H.); juergen.boehner@uni-hamburg.de (J.B.)
\* Correspondence: juliana.freitas.santos@uni-hamburg.com; Tel.: +49-40-42838-7826

**Abstract:** The identification of the biogeophysical effects due to land-use, land-cover, and land-management changes (LULCC) is yet to be clearly understood. A range of factors, such as the inclusion of an interactive ocean model component, representation of land management, transient LULCC, and accountability for atmospheric feedback, potentially shifts how models may detect the impacts of the land surface on the climate system. Previous studies on the biogeophysical effects of LULCC in South Asia have either neglected one of those factors or are single model results. Therefore, we analyzed the outputs from 11 models, participants of the Coupled Model Intercomparison Project in its Sixth Phase (CMIP6), which derived from experiments with and without LULCC and compared the two simulations with respect to changes in near-surface temperature and total precipitation means. The CMIP6 simulations, to a certain extent, accounted for the elements previously overlooked. We examined the grid cells that robustly indicated a climatic impact from LULCC. Additionally, we investigated the atmospheric feedback and the dominant fluxes with their associated land surface variables involved in the changes in temperature and precipitation. Our results indicated that the biogeophysical effects from LULCC favored surface net cooling and surface net drying over the robust areas at all seasons. The surface net cooling was strongly influenced by the decrease in available energy and the increase in latent heat and total evapotranspiration. Surface net drying was highly promoted by local hydrological processes, especially in areas outside the monsoon core. The study also revealed that non-local sources might influence precipitation in some parts of South Asia, although this was inconclusive. Our research presented similar results to previous studies but with different magnitudes, which highlighted the added value of CMIP6-GCMs simulations but also their pitfalls.

**Keywords:** land use; land management; land-use modeling; biogeophysical effects; South Asia; CMIP6

## 1. Introduction

Land Use, Land Cover, and Land Management Changes, hereafter LULCC, can influence the climate through carbon, energy, and moisture fluxes' exchange. Changes in land surface properties that modify the energy and moisture fluxes are called biogeophysical (BGP) effects. BGP effects are often a neglected element in international climate policies, primarily due to high uncertainties, scale dependency, and the contentious issue of global negligibility [1–4]. However, at regional and local scales, BGP effects are notorious [5] and could be eight times larger than global effects on, for example, lower tropospheric air temperatures [6]. Hence, the regional identification of BGP effects is indispensable, especially in vulnerable areas such as South Asia. The vulnerabilities faced by the communities living in this region are not only due to the high dependence on the monsoon regime for agricultural activities but also due to insecurity with regard to resource and risk management [7,8]. Accordingly, we thoroughly investigate the BGP effects in South Asia.

A comprehensive assessment of LULCC-induced BGP effects on climate variables, such as near-surface temperature, should incorporate local and potential non-local effects.

Local effects include both direct and indirect effects [9]. Local direct effects are represented by an immediate response of energy and moisture fluxes to a certain change in land surface characteristics. The indirect effects are the induced response to initiated local direct effects [9], such as the further increase in latent heat release of 27.8% due to the increase in precipitation intensity in monsoon regions [10]. The non-local effects express the potential feedback from LULCC elsewhere such as via teleconnections. The non-local effects combined with the indirect effects convey the atmospheric feedback [11]. This feedback can contribute to more than 75% of surface temperature change over 28% of an area with perturbed land [12], but these effects can also incorporate weather noise [13]. Thus, careful examination of the potential atmospheric feedback is necessary.

Further requirements for a thorough investigation of BGP effects include land management or land-use intensification. Ref. [14] confirmed that due to Earth's limited terrestrial space, land-use intensification is a more conceivable pathway of change than land cover expansion. Ref. [15] highlighted the importance of considering land management (irrigation) as a potential climate-effective mitigator by demonstrating its capacity to minimize regional hot extremes. Another aspect suggested that including annually updated changes or transient LULCC could reduce biases and significantly improve interannual temperature and precipitation variability representations [16–18]. In comparison to a prescribed Sea Surface Temperatures (SSTs) experiment, an interactive ocean model potentially enhances or dampens the magnitude of BGP effects, although the sign of change may remain the same [19,20]. Clift and Plumb [21] noted that the temperatures in the Indian Ocean are vastly dependent on land, ocean, and atmosphere feedback through surface winds. Consequently, an analysis of BGP effects should be able to: (a) account for atmospheric feedback, (b) include land management as an integral part of the land-use dataset, (c) use a transient LULCC dataset where interannual transitions are represented, and (d) have an interactive ocean component.

In most South Asian countries, agricultural expansion and intensification largely increased during the 20th century [22]. All these changes have been associated with the decline of forest cover and other naturally vegetated landscapes, especially during the last few decades [23–28]. Vegetation degradation from crop plantations or shrubland was also observed with an increasing area of barren land [29]. Despite an overall decrease in natural vegetated and forested landscapes, there were many areas in Nepal, Bhutan, or Sikkim experiencing an increase in forest cover due to reforestation programs, effective forest conservation policies, and enforcement of community forestry and sustainable agroforestry systems [30–32]. Regarding land management, irrigation has increased steadily since the 1950s in South Asia, markedly in India [33].

Many studies have assessed the BGP effects in South Asian countries through various methods and different model setups. A few studies looked at the general effect of LULCC without including irrigation, transient land use, and an interactive ocean [23,25,34–36], whereas others retrieved their results from fully coupled models [6,24,37–39]. Further studies looked at the specific effects of irrigation on temperature changes but principally on rainfall trends. These studies repeatedly pointed out the weakening of summer monsoon rainfall due to LULCC, which is partially associated with the land surface cooling reducing thermal gradient between land and ocean [8,35,38,40–42] but also linked to local and remote hydrological responses [24,36,43,44]. Irrigation in South Asia is responsible for decreasing land surface temperature through evaporative cooling [45] and weakening of monsoonal rainfall [46], but it also decreases rainfall in seasons other than summer [42,47]. One singular study differentiated the effects of other-LULCC types from only irrigation and found that these two elements have opposing influences [40]. A recent study by [48] compared three CMIP6-GCMs with active irrigation to those without it and they found cooling trends mostly from the models with irrigation, especially in South Asia. Nevertheless, all these studies did not capture the full range of BGP-LULCC effects or were the result of a single-model analysis. Single-model analyses can partially attribute the changes in temperature and rainfall to model-dependent interactions [25]. By doing so, one can suspect that the

quantification and analysis of BGP effects in South Asia were incomplete or potentially ambiguous through underestimation or overestimation of these effects.

Therefore, our research addresses the yet incomplete understanding of BGP effects through a new set of data that (a) make use of transient annual land-use changes; (b) includes, partially, land management (irrigation); (c) integrates indirect effects; and (d) employs ocean coupling. Although the new dataset was previously explored [49], it was not assessed for the already given aspects. We utilized a subset of fully coupled global climate models (GCMs) from the Coupled Model Intercomparison Project in its Sixth Phase, CMIP6 [50], and its two endorsed activities, the Land Use Model Intercomparison Project, LUMIP [51], and the Land Surface, Snow and Soil moisture Model Intercomparison, LS3MIP [52] to bridge this gap.

We analyzed CMIP6-GCMs outputs from two experiments, with and without LULCC, since the simulations considered, to a certain extent, the mentioned requirements (aspects a to d). We looked at the BGP effects from LULCC in South Asia by answering three research questions: (i) How did the total BGP effects from LULCC affect the annual and seasonal means of the near-surface temperature at 2 m (TAS) and total precipitation (PR)? (ii) How strongly does the atmospheric feedback contribute to the total BGP-LULCC signal? (iii) What is/are the most dominant flux(es) of the BGP-LULCC effect on TAS and PR changes?

## 2. Materials and Methods

### 2.1. Study Area

#### 2.1.1. Climate System

The South Asia region generally involves seven countries: India, Nepal, Pakistan, Bangladesh, Sri Lanka, Maldives, and Bhutan (Figure 1). The region is marked by an important and noticeable change in the seasonal pattern of rainfall, characterized by the wet period of the summer monsoon, associated with up to 80–90% of the annual precipitation, and dry winters [21]. Seasonal rainfall changes are the result of seasonal reversals of atmospheric circulation, which are triggered by the differences in temperatures between both hemispheres and between the land and adjacent oceans [8]. The changes in rainfall patterns are influenced by seasonal changes in the low-level winds, a south-westerly jet coming from the Arabian Sea and the Bay of Bengal passing through the Indo-Gangetic plain [53]; they are also influenced by changes in the upper-tropospheric level winds, the subtropical easterly jet and the Tibetan Anticyclone [53].

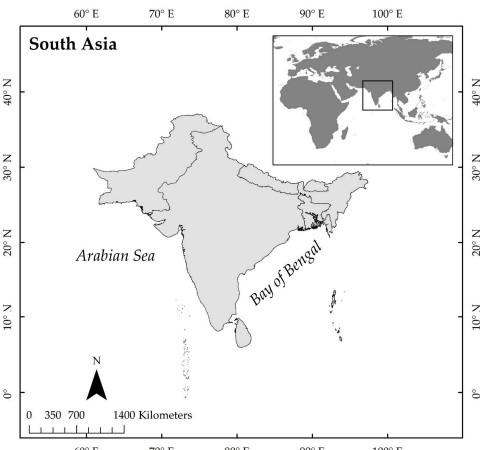

**Figure 1.** South Asian countries included in the analysis.

The development of a pronounced low-pressure heat system over terrestrial South Asia initiates the shift of the intertropical convergence zone (ITCZ), during northern summer towards the north of the Equator at approximately 20° to 25° degrees north, and the following mechanisms involved in the main monsoon season [54]. The northward shifting

of the ITCZ creates a thermal contrast of the boundary layer moist static energy between land and sea [21]. The moist static energy and moist convection, as summer progresses, shift from the adjacent ocean towards the continent and bring rainfall over the South Asian region. The rainfall maxima are located around the ITCZ zone, along the coastal areas, and where orography is markedly present, such as in the western Ghats and the foothills of the Himalayas [21]. The monsoon onset is then characterized by a sudden and very rapid change in atmospheric circulation, where warm and moist air gradually progresses toward the western and northern areas of the region. However, precipitation in this season is not intermittent and it often weakens as it gets further inland [21].

Undoubtedly, the South Asia monsoon system is a fully coupled ocean-land-atmosphere system driven by dynamic and thermodynamic processes, which are influenced by large-scale circulation, mesoscale convective systems, and the fixed orography of the Himalayas, Tibetan Plateau, and the western Ghats [55]. All of these factors dictate the timing, intensity, and duration of the monsoon season, which determine the precipitation pattern over the region [23].

### 2.1.2. Natural Vegetation

The region is covered by a great diversity of landscapes and ecosystems and a large range of topography. India, for example, is covered by moist tropical forests on the west coast, eastern Himalayas, and west of Bengal, dry forests towards the northwest areas, around the hot Thar desert, and the southeast coasts of Tamil Nadu, as well as montane subtropical, temperate forests and subalpine forest in the Himalayan areas [54]. Pakistan (nowadays Pakistan and Bangladesh) is dominated in the lowlands by a tropical thorn, savanna-like, forest, by dry subtropical forests along the border with Afghanistan and in the Himalayan foothills, and towards the north covered by more diverse vegetation with the presence of Himalayan temperate and subalpine forest and alpine scrub and meadows [56].

### 2.1.3. Agriculture Patterns

The precipitation pattern in South Asia is essential to agricultural productivity and the economy of these countries, where the high regional variability of rainfall, the marked seasonal changes, and the somewhat irregular rainfall timing and intensity might put limits on crop production.

There are two cropping seasons in South Asia, the main season during summer, commonly known as Kharif, and the winter crops, Rabi [57]. The latter season is heavily supported by the use of irrigation, as the season is generally dry [57]. Irrigation, which covers 40% of the total area used for agriculture [58], is inevitable considering that rainfall is seasonal, irregular and unevenly distributed [54].

### 2.2. Overview of CMIP6-GCMs
#### 2.2.1. Simulations and Models

We used two simulation experiments, *historical* (from CMIP6) and *hist-noLu* (from LU-MIP/LS3MIP) in the analysis of LULCC-induced BGP effects. The *historical* and *hist-noLu* experiments represented simulations with LULCC and those without LULCC, respectively. These experiments were identical in their settings (prescribed atmospheric $CO_2$ concentrations, all forcing, GHGs concentration, emission of short-lived species, global gridded land use, solar, volcanoes, aerosols, and modeled SSTs and sea ice concentrations) with the exception that land use in *hist-noLu* was kept constant at the 1850 pre-industrial level. Hereby, we could compare two almost identical experiments, excepting the land-use feature, and assign LULCC as the cause of the identified differences. The treatment of constant land use can differ between models, where some models are driven by annual land-use states and others by land-use transition (to another category) rates [51].

From each experiment (*historical* and *hist-noLu*), we considered the first realization of 11 fully coupled GCMs participating in the CMIP6 and LUMIP/LS3MIP listed in Appendix A (Table A1). The analysis was based on one ensemble member of each model,

taking into consideration that the 11 different models had a distinct number of realizations, but all models had at least one. Only taking one realization might have complicated the distinction of the real climate signal from random noise. However, we expected that accessing changes between similar experimental setups would decrease random variability within a model. Besides, we investigated the mean over a long period, which could minimize some of these uncertainties

The *historical* and *hist-noLu* simulations began in 1850 and ran until 2014, from which we investigated the long-term annual and seasonal means between 1950 to 2014. We chose this period because of the steady increase in irrigation observed since 1950. The performance of the CMIP6-GCMs in South Asia was evaluated by several studies [30,59–62], hence, we did not emphasize assessing the model skill but discussed our results given the reported model skills. In general, CMIP6-GCMs showed cold and wet biases over most of South Asia.

### 2.2.2. Assessed Variables

We examined two main climate variables, TAS and PR, and another 15 variables representing the elements participating in the energy balance and moisture budgets. The latter variables were included in the assessment of the potential driving forces of LULCC-induced BGP effects. Table A2, in Appendix A, discloses a complete list of these variables and their brief definitions. All the variables were retrieved from each experiment, *historical* and *hist-noLu*, and subtracted one from the other to derive a change.

To better interpret the results of TAS and PR changes, we analyzed the modifications that were implemented to land-use and land-cover change for each model. The land-use change information was taken from the variable 'fracLut'. The fractional coverage of plant functional types (PFTs) was normally integrated into land-use tiles and distributed over a single grid cell. The fracLut variable distributed land use into four main classes: "Primary and Secondary vegetation" (disturbed and undisturbed natural vegetation, bare ground, and wetlands), "Cropland", "Pasture" and "Urban". Out of the 11 models, CanESM5 and BCC did not provide their land-use fraction information. The bare soil fractions were retrieved from the 'baresoilFrac' files and represented the percentage of an entire grid cell that was covered by bare soil.

All variables were single files as model outputs and were analyzed for their annual and seasonal means. The seasons were represented by winter (DJF), pre-monsoon (MAM), summer (JJAS), and post-monsoon (ON). The CMIP6 dataset was acquired from the CMIP6 database website at the DKRZ server "https://esgf-data.dkrz.de/search/cmip6-dkrz/ (accessed on 21 November 2022)". Further information on the above experiments and land use can be found in the CMIP6, LUMIP and LS3MIP v1.0 reports [50–52].

Before the assessment, TAS files were re-gridded to a common horizontal resolution of $1.9° \times 1.9°$ using bilinear interpolation. The smooth nature of temperature data is fittingly performed with bilinear interpolation [55,59]. PR was resampled using first-order conservative remapping, which was suitable for discontinuous data, especially in fully coupled climate models [63]. The fluxes were also accordingly re-gridded when compared to TAS or PR. The land-use variable was re-gridded using nearest-neighbor interpolation, which was appropriated for categorical data.

### 2.3. Procedures

#### 2.3.1. Robustness and Statistical Significance

We produced robust results on account of two measures: (1) Statistical significance of the differences between the model means of the paired-experiments (*historical* and *hist-noLu*) applied for each model; (2) majority model agreement on the direction of change where at least 60% of the models (seven out of eleven models) agreed on the same sign of change.

Time series of climate data are serially dependent, thus not accounting for autocorrelation imposes a challenge to any statistical test on the significance of a change. For this, we used the modified Student's *t*-test proposed by [64] given that the test accounted for

autocorrelation. The test has also been proven to be moderately robust against deviations from a normal distribution [65]. When a particular grid cell passed the modified two-tailed *t*-test at a 95% confidence level (i.e., *p*-value < 0.05) and the majority of individual models agreed on the direction of change, this grid cell was considered robust. The robust grid cells within the South Asia region were the cells we further investigated to explain how TAS and PR respond to LULCC and how tightly these responses correlated to certain fluxes or land surface variables changes.

2.3.2. Statistical Correlations and the Atmospheric Feedback

We estimated the relationship between the land surface variables and fluxes (independent variables) to TAS or PR changes (dependent variables) by applying simple linear regression. We looked at how fittingly the independent variables explained the variations in the dependent variables through $R^2$. The $R^2$ values were not expected to be very high because we were including sample data from different models. Different models translate into different ways of modeling and hence, heterogeneous results. Some models might be more sensitive to soil moisture changes, for example, than others. The data input as the dependent and independent variables consisted of anomalies and were de-trended. Values from the GFDL-ESM4 model were excluded from the analysis of fluxes considering that this model did not output all variables of our interest.

The quantification of the atmospheric feedback required further inclusion of experiments, which did not account for atmospheric feedback, the so-called offline simulations. We used land-only offline experiments, *land-hist*, and *land-noLu*, in order to disentangle the direct effects from the atmospheric feedback of LULCC on TAS. The offline experiments were equivalent to *historical* and *hist-noLu* experiments respectively, however, no atmospheric feedback was allowed. The calculation of the atmospheric feedback strength was performed by subtracting the fully coupled simulations from the offline simulations, following [12].

Figure 2 summarizes the methodological steps of the study. In the results section, the comparison between the two experimental setups is referred to as "changes", and the intermodel spread and particularities of a model are referred to as "differences". In Section 3.1 of the results we analyze the LULCC simulated by the 11 models and compared them with real trends of LULCC in South Asia. Sections 3.2–3.4 explore the robust changes in TAS, the dominant fluxes, and land surface variables important to these changes, and the atmospheric feedback, respectively. Sections 3.5 and 3.6 present the PR changes and the fluxes and variables associated with changes in the moisture budget leading to the potential modifications in PR. Finally, the results are reasoned in the discussion section where explanations of the identified changes in TAS and PR are given.

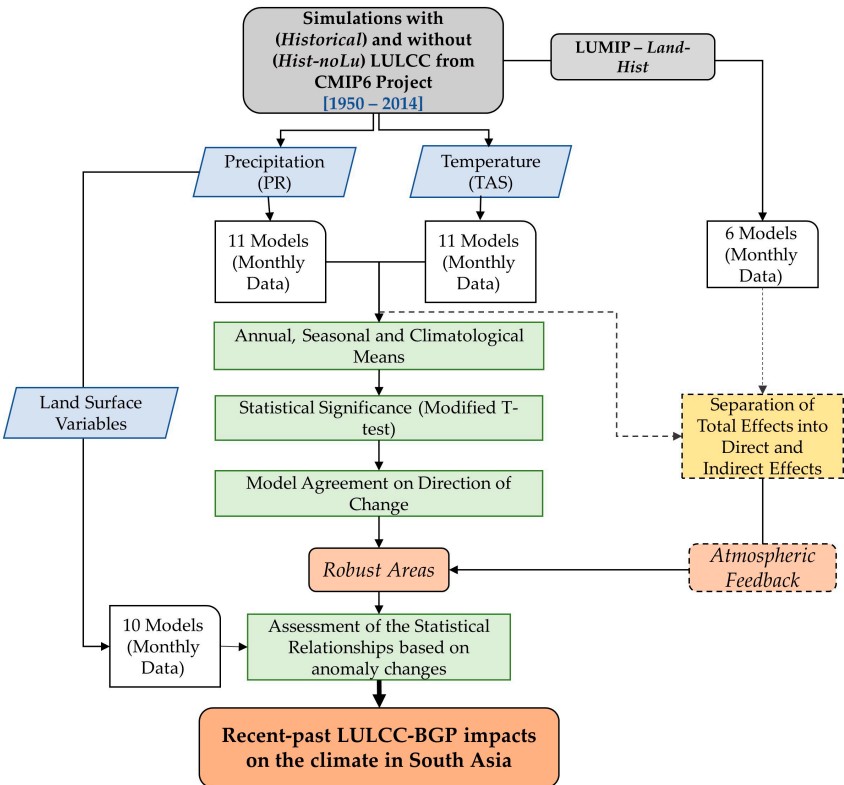

**Figure 2.** Methodological steps to the full-BGP effects of LULCC in South Asia based on the analysis of CMIP6-GCMs.

## 3. Results

### 3.1. LULCC and the Differences among Models in South Asia

The measure of LULCC was seen through changes in vegetation structure, primarily through the Leaf Area Index (LAI), and natural vegetation distribution by fractional coverage of PFTs incorporated into fracLut. The outcome from fracLut revealed that in the robust grid cells of both TAS and PR, LULCC was configured as a mean increase in cropland and pastures at the expense of primary and secondary vegetation (Figure 3).

The largest decreases in primary and secondary vegetation were exhibited by GFDL (−7.45%) and MIROC (−9.86%) models, and the peaks of cropland increase were simulated by CESM2 (+8.16%) and MPI (+7.67%). Pasture increase was not very representative for all but a few models, especially MIROC, which simulated an +8.57% increase (Figure 4). Besides, most models did not have a representation of pastures but rather of grasslands, often being integrated into primary and secondary vegetation. The values on bare areas were not available for all models, and most models simulated a maximum decrease of −3.24% (CMCC, CESM2) but one model showed an increase of +5.05% (UKESM).

According to studies on historical changes in land use and land cover in South Asia [22–28], forest and natural grasslands have decreased between 1950 and 2000 on account of cropland expansion (somewhat limited) and agricultural intensification (most prominent), which agrees with the changes in land cover simulated by most CMIP6 models included here. The spatial pattern of changes shown by most models exhibited a higher average increase in cropland over central India (Figure 3). One particular model (UKESM1), however, presented no changes in cropland over central India and western Pakistan. Central India is among the areas where cropland has increased or intensified the most since the 1950s [43]. Moreover, the prevailing decrease in bare areas over most parts of South Asia does not coincide with the increase in land degradation experienced in the past century [29].

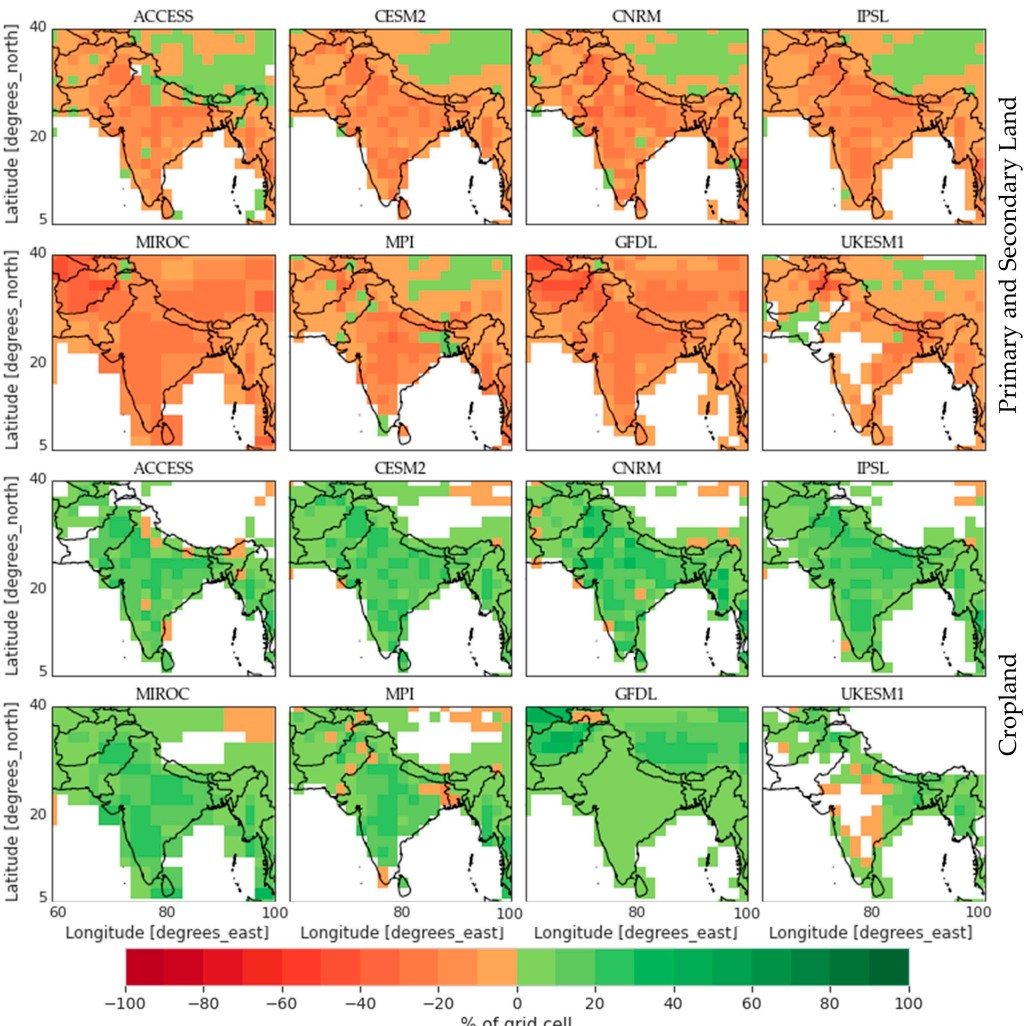

**Figure 3.** Spatial distribution of the LULCC retrieved from fracLut variable. Negative values indicate a loss and positive ones are a gain in the respective grid cells. Maps display the mean annual values between 1950 to 2014.

The slight differences observed among the models on how they included LULCC in their experiments depended mostly on the adopted land-use forcing dataset and how models translated land-use information into their PFTs' classes and represented them back into land cover classes. Since all models claimed to use the same land-use forcing dataset (LUH2), the differences were the results of varying ways that these models integrated vegetation and crop types into their land surface models. Models with large discrepancies from the LUH2 consistently (Figure S1) overestimated the increase in primary and secondary vegetation. Assuming that LUH2 already had some embedded uncertainties due to coarse spatial resolution and the accuracy of the datasets incorporated to derive LUH2, adding deviations to it further restricted the model's capacity to accurately represent reality.

Another crucial aspect of land surface characteristics was the change in vegetation structure. An important factor of vegetation structure is the leaf area index (LAI), a ratio between plant foliage cover to plant ground cover. A reduction in LAI (combined with tree/plant height) reduces surface roughness and increases aerodynamic resistance, resulting in lower efficiency to transfer turbulent fluxes. Furthermore, with LAI decrease, less water can be intercepted by tree canopies and thus become available for evaporation. Having all that in mind, we performed a simple analysis of the LAI annual cycle and spatial distribution from the examined CMIP6 models using the model output variable, LAI, from their *historical* simulations. We found that LAI peaked for most models around

September–October (transition of summer to post-monsoon season), which is corroborated by [49] and represented well the LAI cycle over South Asia (Figure 5). However, spatially (Figure 6) LAI does not coincide with the changes in vegetation distribution provided by the fracLut variable (Figure 3) for some models.

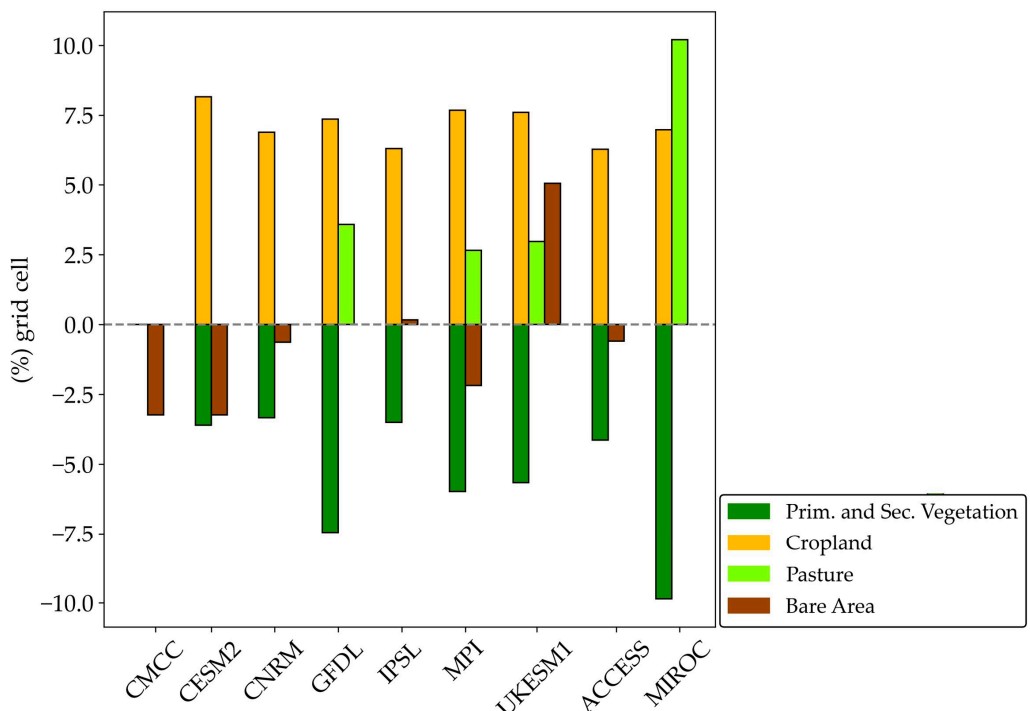

**Figure 4.** Quantification of LULCC (*historical—hist-noLu*). Values represent the mean annual, between 1950 to 2014, averaged over the robust grid cells for TAS. Tas and PR values are very similar. More details on bare area changes, which fluctuate between the seasons, are available in Figure S2 in the supplementary material.

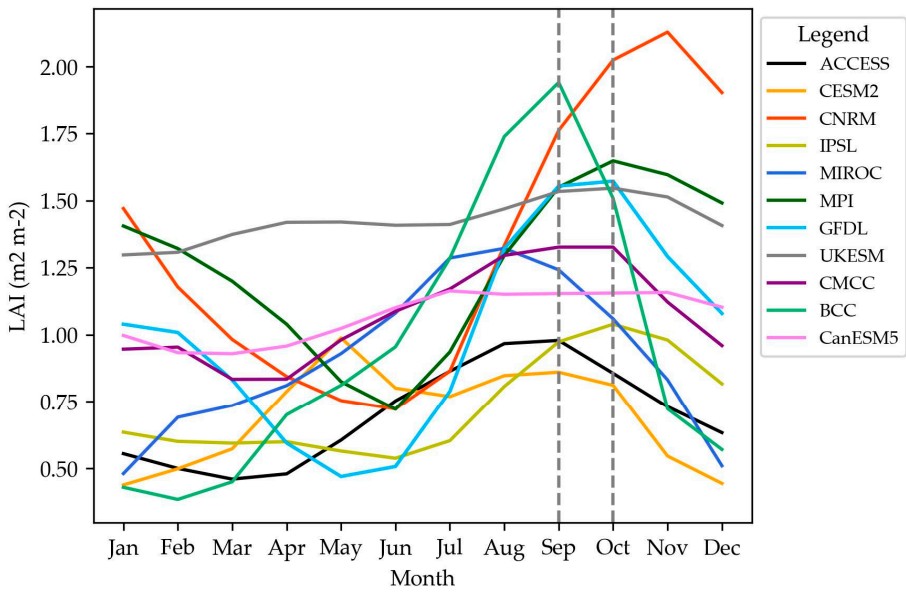

**Figure 5.** LAI climatologies from 1950 to 2014 in South Asia. The dotted gray lines indicate the two months when LAI peaks for most models.

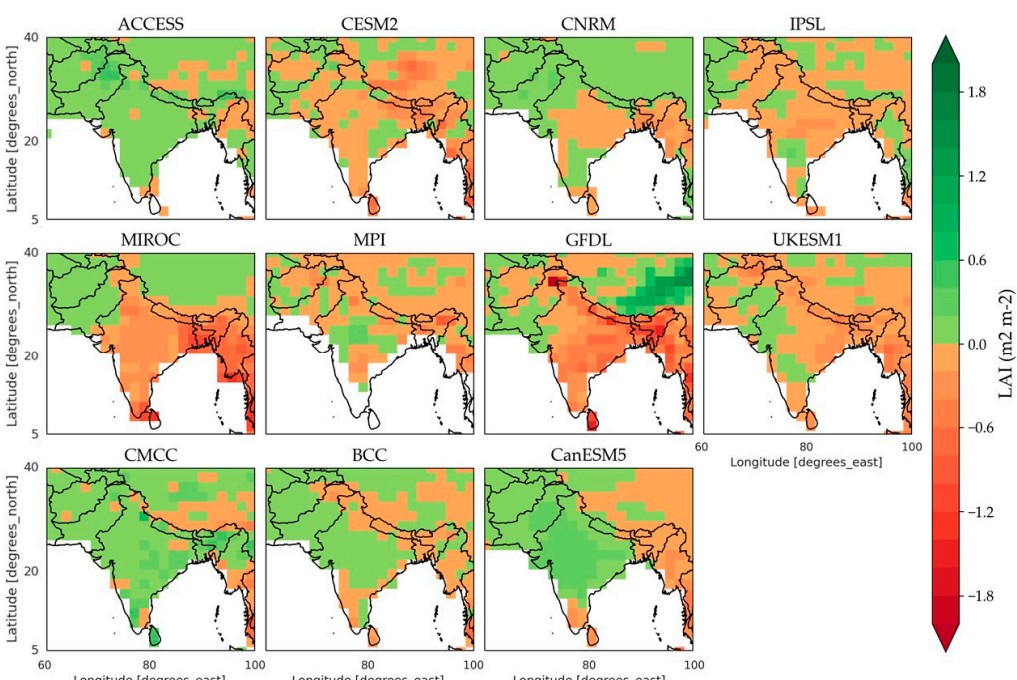

**Figure 6.** Spatial distribution of LAI changes (*historical—hist-noLu*) for each individual model. The color code follows the same logic as in Figure 3.

The main reason for the discrepancy between LAI output and fracLut information lay in the many ways land surface models integrated the PFTs into their land-use cover tiles and whether LAI was prescribed or not. As an example, the CMCC model had only two types of land-use tile, urban and vegetated, where the vegetated land-use tile would remain the same throughout time in both experiments, *historical* and *hist-noLu*. Vegetation changes would only occur within the "vegetated" tile for the PFTs distribution, so LAI might change regardless of the changes in the "vegetated" cover. The ACCESS, CMCC, CanESM5, BCC, and partially the CNRM models showed an increase in LAI despite the clear decrease in primary and secondary vegetation (Figure 3). The models depicting values of LAI corresponding to its vegetation fraction changes were the MIROC, CESM2, GFDL, IPSL, UKESM, and partially the CNRM and MPI, all values extracted from annual averages. Additionally, LAI changes were not only reflected in LULCC but could also be a response to changes in carbon dioxide and the overall changing climate. Another relevant facet was the varying ways land surface models resolved LAI, being prescribed or prognostic. If a model prescribed LAI, then LAI changes would be dissociated from the surface climate changes experienced by the same model. Moreover, ref. [49] highlighted that LAI was overestimated by all CMIP6 models and for most places in the world, with overestimation mainly deriving from non-forested cover. Finally, the interannual cycle of LAI in a model is highly dependent on the seasonal cycle of crops and the length of the growing season resolved in the model. The differences in how to retrieve changes in vegetation properties will ultimately influence the LULCC forcing signal, which might favor a certain direction of change.

### 3.2. Model Responses to TAS Changes

Most models (at least 7 out 11) agreed on area average net cooling favored by LULCC at mean annual and in all four seasons, which showed no clear seasonal pattern. The spatial distribution of the robust grid cells slightly differed depending on which seasonal mean is considered (Figure 7).

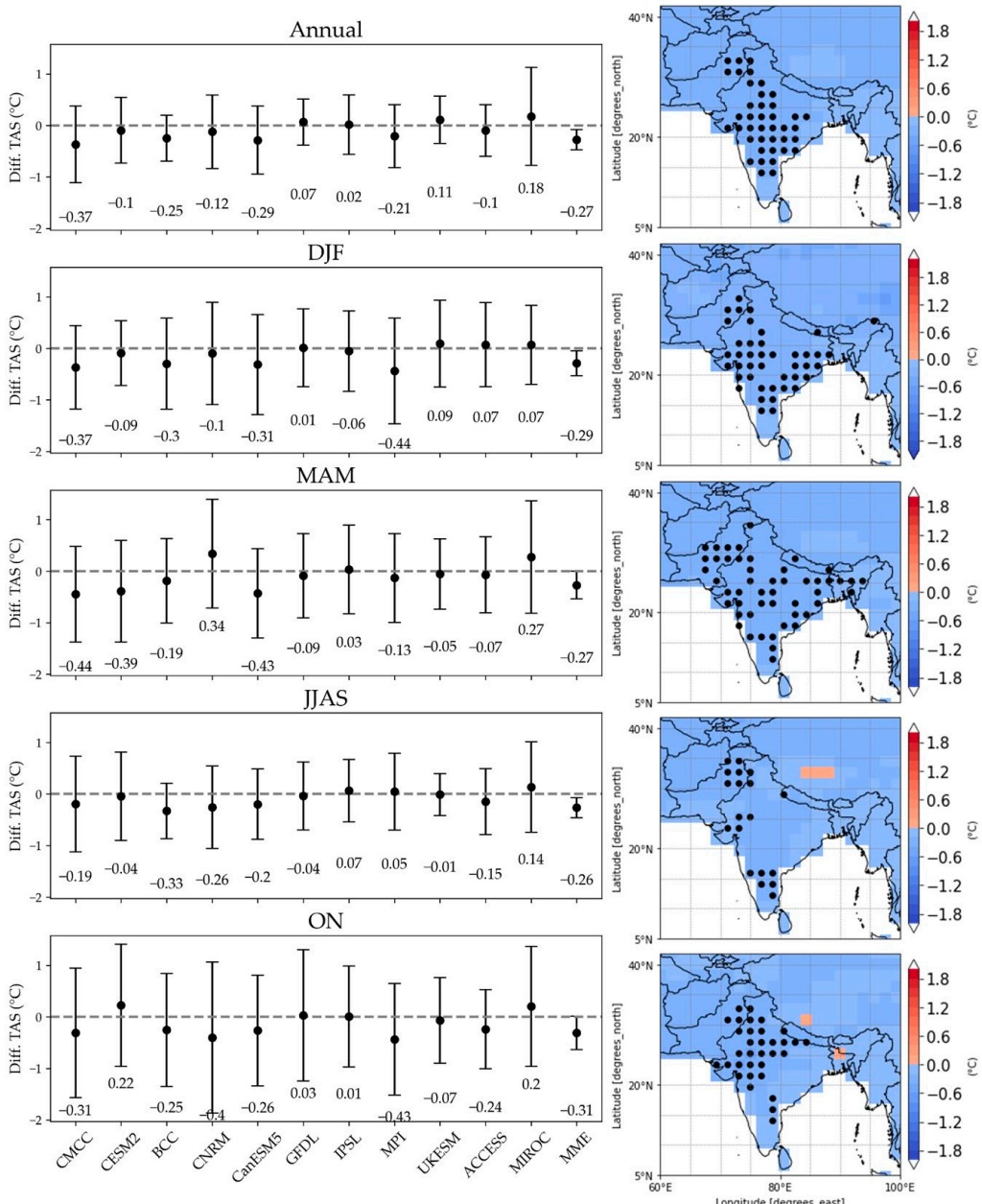

**Figure 7.** On the right-hand side, the spatial distribution of the multimodel mean changes in TAS (*historical* minus *hist-noLu*) at annual and seasonal temporal scales. The blue colors illustrate cooling and the robust areas are marked by a black dot. On the left-hand side, the CMIP6-GCMs model spread (standard deviations and model-mean) over robust areas for the respective temporal scale.

However, there were parts of the region with consistently robust results throughout the years and the seasons, such as the eastern part of Pakistan, and the central-northwestern and southern tip of India. Winter, pre-monsoon, and post-monsoon seasons were the seasons with the strongest net cooling changes with mean values of −0.44 °C (MPI), −0.44 °C (CMCC), and −0.43 °C (MPI), respectively (Figure 7). The net cooling resulting from the influences of LULCC changes in South Asia was spatially dominant during the pre-monsoon season. The extent of temperature change during summer was the smallest of all seasons. Generally, small-scale vegetation changes did not influence the main monsoon season as much due to the dominance of the ocean forcing during this wet season [38].

### 3.3. Changes in the Energy Fluxes

The surface energy balance equation (not shown) consists of fluxes influencing the radiative and the non-radiative sides of energy exchanges. We looked at 15 different variables to examine their influences on the radiative and non-radiative exchange of energy between land and the atmosphere.

#### 3.3.1. Radiative Fluxes

The radiative side of the surface energy balance is broadly represented by the net radiation (Qnet) that is net shortwave radiation (net SW) summed to net longwave radiation (net LW). On the one hand, Qnet does not display high $R^2$ values (Figure 8) and is typically influenced by cloud cover (Cc). Therefore, Qnet and TAS changes are likely not to be linearly correlated. Qnet responses were highly subject to other factors, such as cloud cover (Cc), which in turn might shape how changes in Qnet varied with TAS changes. On the other hand, the available energy (Qa) was the dominant flux in shaping changes in TAS as Qa was the variable with the highest $R^2$ values (over 60%) for most temporal scales (Figure 8). The regression coefficient between changes in Qa relative to changes in TAS was positive in all seasons and the annual mean (Table 1). Qa corresponds to longwave downwelling flux (LWd) plus net shortwave radiation (net SW). A decrease in Qa means that less energy is available to warm up the land when the energy used as a source of heat reduces, hence cooling. Connected to a decrease in Qa, the net longwave radiation (net LW) was the variable with the second highest $R^2$ values among the radiative fluxes, especially in summer, when 64.03% of TAS changes could be explained by changes in net LW (Figure 8). The regression between TAS and net LW was negative, which indicated that a decrease in TAS was associated with an increase in net LW. The increase in net LW simply meant that more energy left the system in the form of longwave radiation in simulations with LULCC. This pattern corroborated with the increase in latent heat (LH), total evapotranspiration (Esurf), and evapotranspiration from soils (Esoil) simulated by most models (mean values not shown). Moreover, Cc appeared to be generally uncorrelated with TAS changes, except during the summer season. As mentioned earlier, $R^2$ values for NetLW explained a little over 60% of the variations in TAS in the summer season and Cc was responsible for about 57% of TAS changes. Cc was negatively correlated to TAS, which indicated that the increase in cloud cover had a cooling effect on the surface by probably blocking some of the shortwave radiation to reach the surface.

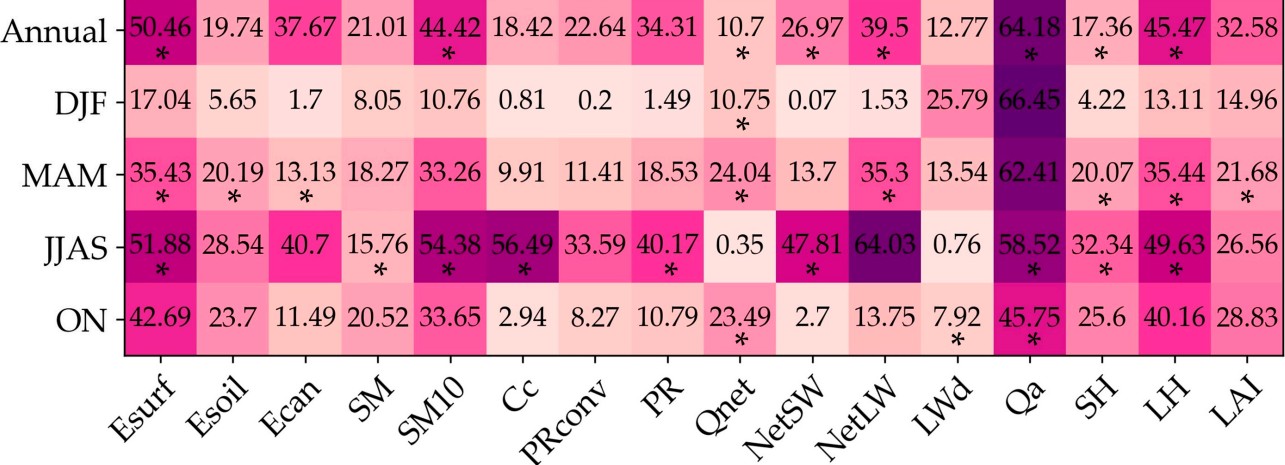

**Figure 8.** Heatmap with $R^2$ values of the linear relationships between TAS changes and the fluxes with their associated land surface variables. The strength of the relationship is characterized by the intensity of the shaded color, where darker shades denote stronger $R^2$ values. Values marked with an asterisk are significant at a 95% confidence level.

**Table 1.** The regression coefficients of the linear relationships described in Figure 8.

|         | Esurf  | Esoil  | Ecan   | SM    | SM10  | Cc     | PRconv | PR     | Qnet   | NetSW  | NetLW  | LWd    | Qa    | SH    | LH     | LAI    |
|---------|--------|--------|--------|-------|-------|--------|--------|--------|--------|--------|--------|--------|-------|-------|--------|--------|
| Annual  | −3.19  | −1.75  | −7.80  | 0.00  | 0.00  | −0.11  | −0.82  | −0.94  | −0.07  | 0.11   | −0.11  | 0.07   | 0.11  | 0.06  | −0.10  | −2.89  |
| DJF     | −1.16  | 2.91   | 17.46  | 0.00  | 0.00  | 0.02   | −1.29  | −0.21  | −0.09  | 0.00   | 0.00   | 0.05   | 0.14  | 0.07  | −0.01  | 0.08   |
| MAM     | −2.39  | −4.80  | −4.31  | 0.00  | 0.00  | −0.12  | −1.73  | −1.17  | −0.11  | 0.07   | −0.09  | −0.02  | 0.14  | 0.12  | −0.10  | −5.55  |
| JJAS    | −1.83  | −1.20  | −8.85  | 0.00  | 0.00  | −0.14  | −0.74  | −0.59  | −0.01  | 0.10   | −0.09  | −0.01  | 0.08  | 0.07  | −0.06  | −2.12  |
| ON      | −3.33  | −4.14  | −6.90  | 0.00  | 0.00  | −0.02  | 1.63   | 0.82   | −0.09  | 0.01   | −0.02  | 0.08   | 0.11  | 0.09  | −0.11  | −2.78  |

The values for SM and SM10 are numerically represented by 'zero' as they are very small.

Qnet, Qa and their related fluxes return only the radiative effects from LULCC. These effects may not alone contribute to surface net cooling simulated by models since BGP effects are often mingled responses, which can easily offset each other. The non-radiative forces component essentially controls energy through the partition of Qa into sensible heat (SH) and latent heat (LH).

### 3.3.2. Non-Radiative Fluxes

The flux with the highest $R^2$ values among the non-radiative fluxes and their associated variables is latent heat (Figure 8). The latent heat is the heat taken from the surface during evapotranspiration process. Consequentially, Esurf also displays $R^2$ values similar to or even higher than those from LH. Esurf accounts for Esoil, transpiration from plants, and evaporation from canopies (Ecan). One important aspect of the evapotranspiration process is the amount of water available in the soil to be evaporated. The total soil moisture (SM), but principally the soil moisture of the upper ten centimeters of the soil layer (SM10) had the third highest $R^2$ among the non-radiative fluxes and their associated variables (Figure 8). LAI changes were not strongly associated with TAS changes as they only showed R2 values ranging between 14.96 in winter to 32.58 annually (Figure 8). Moreover, the low $R^2$ values might not necessarily indicate that there was no or very weak correlation but rather that the relationship between the variables is not linear.

### 3.3.3. Winter Responses

Although there was no clear seasonal pattern in TAS changes, flux changes slightly differed during the winter season. One particular characteristic of this season was the lowest $R^2$ values among all temporal scales. Another singularity was that Esoil, Ecan, SM, SM10, and LAI showed opposite signs of regression coefficients (positive) to the same variables in the other seasons (generally negative), see Table 1. Those two aspects strongly suggested that LULCC during winter did not significantly influence the climate and/or that LULCC influences were shaped by the atmospheric feedback and the background climate of the season.

In the next section, we further explore how important the atmospheric feedbacks are to the overall LULCC signal on TAS changes, especially during the winter season.

### 3.4. Atmospheric Feedback

As mentioned previously, LULCC alter the surface properties causing direct local and indirect local impacts with possible remote effects. Atmospheric feedback is understood as the combination of indirect and potential remote responses. We derived the feedback strength by accessing the discrepancies between coupled and offline simulations (see methods). There were six offline simulations available from CMIP6-GCMs models. The following numbers were all median values from CMIP6-GCMs which simulates net cooling responses.

Atmospheric feedbacks were responsible for nearly 23% of the surface cooling over 78% of the robust-grid cells at annual means. These feedbacks were even stronger during the winter season (27.28%) covering an area of nearly 70% of the robust grid cells (Table 2). During pre-monsoon and summer seasons, the non-local responses were the weakest and only impacted in roughly half of the robust grid cells. The $R^2$ measures during summer were the highest for most of the examined variables, which may indicate how variations in TAS

during this season were more of a local nature, even though TAS changes in summer were the smallest (Figure 7). During the post-monsoon season, the interactions of land surface changes with the atmosphere were largely spread over the robust cooling grid cells (89%), controlling about 22% of TAS cooling. The atmospheric feedback could attenuate direct local impacts; consequently, the accountability of the atmospheric feedback is indispensable [11]. It is important to emphasize that the presented results were the median values aiming for simplicity. However, the spread among models should be acknowledged following that the differences between models were quite large for some seasons, which contributed to the overall intermodel spread [9]. The intermodel spread of the feedback strength is shown in the supplementary material (Figure S3).

**Table 2.** Strength of atmospheric feedback (%). The values represent the robust cooling areas and strength refers only to cooling. The unit of area is in % relative to the total robust grid cells respective to the temporal scale.

| Model | Annual | | DJF | | MAM | | JJAS | | ON | |
|---|---|---|---|---|---|---|---|---|---|---|
| | Strength (%) | Area | Strength (%) | Area | Strength (%) | Area | Strength (%) | Area | Strength (%) | Area |
| CESM2 | 21.38 | 23 | 30.12 | 69 | 15.41 | 36 | 19.91 | 32 | | |
| CNRM | 15.43 | 93 | 9.82 | 63 | | | 18.73 | 89 | 17.44 | 91 |
| IPSL | | | 11.05 | 54 | | | | | | |
| UKESM | | | | | 13.60 | 60 | 10.74 | 53 | 16.68 | 88 |
| CMCC | 32.89 | 68 | 29.27 | 70 | 28.71 | 84 | 19.06 | 68 | 27.63 | 79 |
| MPI | 24.10 | 88 | 27.28 | 97 | 21.46 | 54 | 4.62 | 27 | 26.38 | 90 |
| Median-Value | 22.74 | 78 | 27.28 | 69 | 18.43 | 57 | 18.73 | 53 | 21.91 | 89 |

### 3.5. Model Responses to PR Changes

Changes in precipitation had a much larger spread than changes in TAS, resulting in very few grid cells with robust values. Overall models, including the only one with irrigation representation (CESM2), tended to agree on drier parts of South Asia as the main response to LULCC in all seasons, but especially during summer and post-monsoon seasons, over the central-east coast and the southwest tip of India, respectively. The peak changes were (−) 0.88 mm/day (MIROC) and (−) 0.83 mm/day (UKESM) in summer and (−) 0.62 mm/day (CanESM5) in the post-monsoon season (Figure 9). Models also agreed on wetting over one of the parched areas of South Asia, northwest of Pakistan. The wetting agreement only happened during the post-monsoon season with a peak of (+) 0.2 mm/day (CNRM), which was not a very large change (Figure 9). The pre-monsoon season did not show any robust results in the study area for precipitation changes. The percentages of decrease or increase in PR, calculated in mm/year, are shown in Figure 10. This figure suggests that the highest percentages of increase (median values) were seen during the post-monsoon over the wet-agreement areas, followed by dry-agreement during the same season.

### 3.6. Changes in the Elements of the Moisture Budget

Changes in the elements of the moisture budget over dry-agreement areas were more often than not indicated by surface drying that further limits local precipitation recycling. Overall, SM10 and PRconv decreased in all seasons (not shown) and they were the two variables with the highest $R^2$ values in all temporal means (Figure 11). The importance of the relationship between PR changes and SM10 and PRconv suggests that the decrease in PR is linked to the elements participating in the local hydrological cycle. The other related variables, such as Esurf, Esoil, Ecan, and SM, were not strongly associated with changes in PR as they were subjected to other factors, such as temperature. TAS explained at least half of the variations in PR during the summer season (Figure 11).

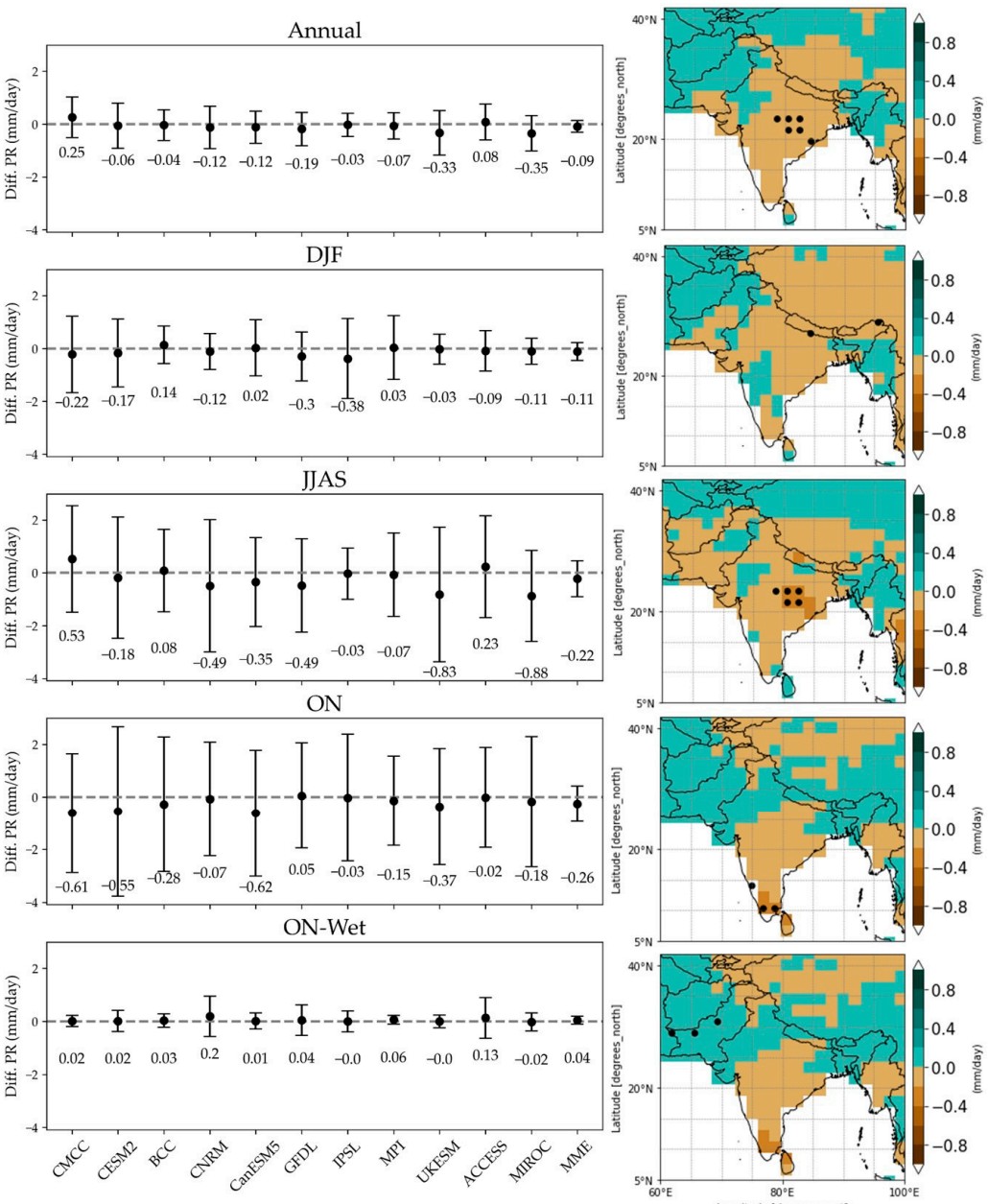

**Figure 9.** Same as Figure 7 but for changes in PR. The brown shades illustrate drying and the green shades show wetting.

Esurf, Esoil, LH allied to SM10 and PRconv were highly responsible for the changes in PR during the post-monsoon season for the wet-agreement areas. The mean values derived from all models in the post-monsoon season (not shown) denoted an overall increase in the availability of water in the soil, higher rates of evapotranspiration, and thus recycled precipitation in the simulations with LULCC. The regression coefficients between land surface variables and PR changes remained positive throughout the seasons and for all variables, except for TAS (Table 3).

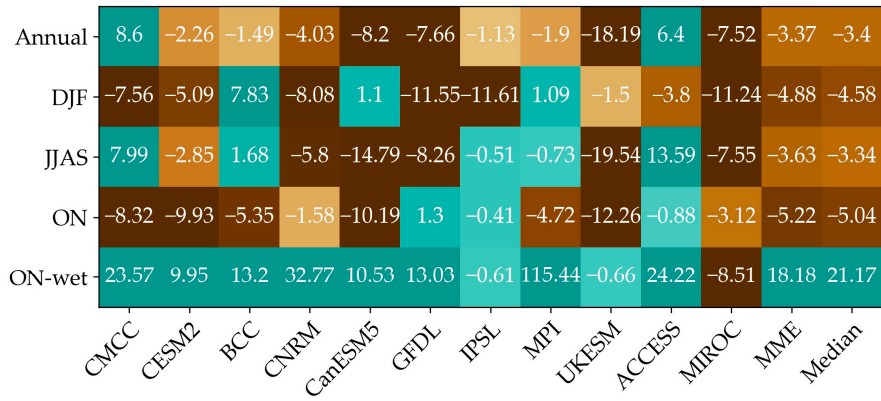

**Figure 10.** Changes in PR (%) based on averages over the robust grid cell at mm/year for the respective seasons and models. The shades of green indicate wet means (differences above zero) and the shades of brown denote dry means (differences below zero). The darker the shade the bigger the difference.

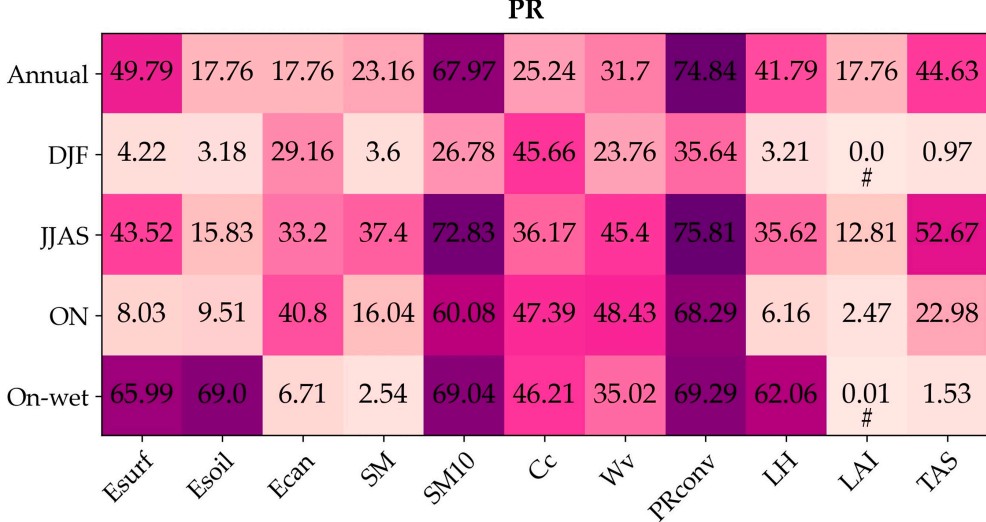

**Figure 11.** Same as Figure 8 but on PR changes. All values are significant at 95% confidence level except the two marked with '#'.

**Table 3.** Same as Table 1 but the values are for the correlation with PR changes.

|        | Esurf | Esoil | Ecan  | SM   | SM10 | Cc   | Wv   | PRconv | LH   | LAI   | TAS   |
|--------|-------|-------|-------|------|------|------|------|--------|------|-------|-------|
| Annual | 1.80  | 1.38  | 1.38  | 0.00 | 0.00 | 0.08 | 0.21 | 1.20   | 0.06 | 1.38  | −0.57 |
| DJF    | 1.28  | 1.28  | 12.69 | 0.00 | 0.00 | 0.11 | 0.45 | 2.51   | 0.04 | −0.01 | −0.10 |
| MAM    | 2.52  | 2.32  | 6.45  | 0.00 | 0.00 | 0.15 | 0.36 | 1.22   | 0.08 | 2.51  | −0.99 |
| JJAS   | 1.78  | 2.22  | 6.41  | 0.00 | 0.00 | 0.15 | 0.40 | 1.21   | 0.05 | 0.58  | −1.52 |
| ON     | 1.77  | 1.95  | 10.98 | 0.00 | 0.00 | 0.03 | 0.11 | 1.77   | 0.06 | 0.04  | −0.04 |

The values for SM and SM10 are numerically represented by 'zero' as they are very small.

The third-order polynomial regression model failed to improve $R^2$ values of the relationships between fluxes and PR changes, as compared to the $R^2$ results from simple linear regression (not shown). Similar to TAS changes, PR changes during winter were not associated with changes in the land surface variables due to LULCC. The winter season was the one with the lowest $R^2$ values.

## 4. Discussion

### 4.1. TAS Changes

The outcomes from TAS changes indicate that the net signal of TAS to LULCC was very sensitive to Qa and associated net LW followed by LH and affiliated Esurf. Ref. [3] pointed out that, contrary to common belief, albedo changes were not the main biogeophysical effect from LULCC detected at the local scale. The surface cooling observed by most models was the result of both evaporative cooling and loss of energy through the reduction of Qnet and Qa.

Although LAI changes are not strongly associated with TAS changes, usually the increment in LAI enhances surface roughness leading to a decrease in wind speed and an increase in the efficiency of both evaporation and convection. Through enhancing evaporative efficiency, the changes in the elements involved in the internal feedback of moisture recycling may ultimately control precipitation, which may further intensify evapotranspiration and surface cooling. Therefore, LAI might not linearly correlate with TAS changes, but LAI changes can influence LH exchange and ultimately TAS changes. Essentially, LH response is rather sensitive to changes in vegetation types and their different parameters and depends on the level of detail in vegetation incorporated by the land surface model. Additionally, as mentioned before (Section 3.1), LAI does not entirely correspond to the land cover distribution changes exhibited by fracLut, nor might the fluxes do so.

LH increase is a clear and common cause of surface cooling; however, LH is not exclusively influenced by the water being evaporated at the interface between land and the atmosphere above. LH is also subjected to other variables such as precipitation. Zeng and Zhang [10] have determined that precipitation in the monsoon regions might partially explain increasing trends in LH. This latter linkage is especially troubling considering that many CMIP6-GCMS in South Asia are showing wet bias for several reasons, with the high sensitivity of convective precipitation in these models being one of them. The direct link between the LULCC signal and some poorly-resolved atmospheric processes in CMIP6-GCMs weakens the reliability of understanding BGP effects in South Asia from these models. Furthermore, convection processes are routinely parameterized in GCMs, making the interpretation of land surface effects on cloud formation rather more complicated. Nonetheless, deforested or degraded areas typically demonstrate a diminished Esurf and LH, as crops and grasses are less efficient than trees in transferring energy to the atmosphere combined with the increase in resistance of air to turbulent transport.

Concerning winter responses, the opposite sign of the regression coefficients observed from Esurf, Esoil, Ecan, LH and LAI in comparison to the same variables in the other seasons suggested that LULCC might weakly interact with the atmosphere above or be shaped by atmospheric feedback and the background climate. Some of the causes which explain the independence of the atmosphere from LULCC during the winter season are the natural vegetation cycle and the atmospheric forces present during this season. Winter is naturally outside of the main vegetation growing period, along with the lower incidence of radiation, making albedo changes from vegetation changes less powerful [4] unless snow-albedo processes are involved. Since snow cover is not present in the CMIP6-GCMs in South Asia (not shown), the snow-albedo effect is excluded, and thus winter cooling from it. The surface cooling observed in this season is likely a response to a decrease in Qnet and Qa, and is partially strengthened by atmospheric feedback and the season's climatic background.

Ref. [6] related the decrease in TAS (0.2 to 0.4 °C) in South Asia to a large decrease in LH, but only during spring and post-monsoon seasons, whereas our studies explored the other monsoon seasons and found a similar proportion of TAS reduction as seen in their study. Although [6] was based on single-model results and did not include transient LU-changes, the model is fully coupled, which potentially influences the similarities in the magnitude of TAS changes between their study and ours.

Admitting that only one model includes irrigation representation (CESM2), most models simulate robust surface net cooling in the region. The robust surface net cooling

is observed by other studies as well [40,45,46], although all have included, in some way, irrigation in their modeling exercise. The distinction here is in the magnitude of the TAS changes from the CMIP6-GCMs and the previous studies. Ref. [40] denoted a cooling of 1–1.5 °C, especially over heavily irrigated areas in South Asia, whereas the range of responses observed here was between 0.01–0.44 °C, including the single-model with a representation of irrigation. Ref, [48] only detected surface net cooling over the northwest portions of the South Asia region, whereas the rest of the region experienced surface net warming. Puma and Cook [45] identified cooling ten times larger, over the northwest parts of the Indian subcontinent, than the surface net cooling reported in this study.

Other studies [24,35,38] simulated an opposite sign of change in TAS to the ones shown here. Ref. [40], when comparing the effect of LULCC with and without irrigation, claimed that LULCC without irrigation induced warming. The responses between models with irrigation representation and without it are not very different in our CMIP6-GCMs analysis. Moreover, our results do not reveal great differences in cooling between CESM2 (the only model with irrigation representation) and the other CMIP6-GCMs as observed in [48]. Surface cooling, in our case, was denoted by the combination of multiple effects causing net cooling. Excepting [48], the studies above did not consider an interactive ocean, or were the result of a single model analysis, or even indicated idealized land-use scenarios. Therefore, those studies contradict the current results and open questions on whether the distinct responses were caused by the high convective sensitivity of CMIP6-GCMs or by omitting one or more of the three aspects already introduced. In addition, the coarse resolution of CMIP6-GCMs complicated the appropriated representation of valley-scale processes that influence the distribution of temperature and precipitation in the highly topographical areas of South Asia, the Himalayan region.

Nonetheless, the CMIP6-GCMs analyses had one distinct advantage over most of the previous studies, which went beyond the robustness of the results (multimodel analysis). The CMIP6-GCMs were fully coupled models and accounted for potential atmospheric feedback acting on top of the local responses.

*4.2. PR Changes*

The changes in the elements participating in the local hydrological cycle were the main contributor to surface drying, due to LULCC, although these changes were not the only cause. SM and PRconv were highly correlated with PR changes, yet the high $R^2$ values were not fully able to explain variations in PR, leaving an open gap. One possible contributor to PR changes, discussed in other studies [8,35,38,40–42], was the large-scale change in temperature over the land surface, which could in principle have affected the thermal sea-land contrast. Lower temperatures over land diminished the thermal contrast between land and sea, eventually causing weaker winter or summer monsoon rainfall and lesser PR in the respective seasons. The trend of TAS indicated that for the majority of the models, temperature over land decreases (not shown). Despite the plausible relationship between TAS and PR changes, the simple assessment of the mean changes in temperature was not enough to conclude with certainty that land-sea thermal contrast was a cause of surface drying. Moreover, how large would the magnitude of the LULCC have to be to significantly impact the thermal forces is unknown. The results illustrated that there were forces beyond the local moisture interactions simultaneously working on drying over South Asia. Perhaps, the drying values presented here could have been even larger, in magnitude, considering the wet biases and the poorly reproduced ocean coupling relationship seen by most CMIP6-GCMs in South Asia.

The spatial pattern of influences corroborated with the local land-atmosphere moisture and the oceanic moisture and high advective humidity transport with the reversal of winds in summer. The summer and post-monsoon robust grid cells were located in areas highly affected by the main monsoon precipitation regime, whereas the grid cells in the winter season and the wet-agreement grid cells were further away or just at the edge of the core monsoon areas. The spatial pattern of the dry-agreement robust grid cells supported the

findings from [36] which demonstrated that in the south, western and southeastern parts of India, regional precipitation recycling ratios were not strong during summer monsoon but might slightly increase in the post-monsoon season. Ref. [42] concluded that changes in land surface properties, such as an increase in moisture input through irrigation, were particularly powerful in South Asia due to stronger soil moisture sensitivity in climatic transition zones.

The post-monsoon season especially favored a high recycling ratio condition because of the increased soil moisture (climatologically wet sea) and increased vegetation cover (mature vegetation), leading to enhanced evapotranspiration. Besides, evapotranspiration is not only influenced by soil moisture and vegetation cover but also by atmospheric conditions. Therefore, in dry regime areas, where most wet-agreement grid cells are located, it was reasonable to observe exacerbated evapotranspiration rates. To draw a parallel, ref. [43] found a much larger increase in evapotranspiration rates in the Indian monsoon belt during the dry season when additional soil moisture (irrigation) was applied. Even so, wet agreement over a particularly dry area of Pakistan (Figure 9), around the Baluchistan region, seemed rather unlikely for two reasons. First, the Normalized Difference Vegetation Index (NDVI) trends in southwest Asia, according to [29], had somewhat decreased in the recent past due to long-term vegetation degradation, especially in croplands and shrublands. Second, the wet bias caused by, among other reasons, the high sensitivity to convection processes, reduced the reliability of the results in an otherwise dry area by either suggesting ambiguous effects or amplified ones. As seen in Figure 10, the percentage of increase in PR over the wet-agreement areas was the largest.

The results from our research corroborate previous studies on surface drying in South Asia due to LULCC [9,24,25,34,38,40–42,44–46]. Ref. [34] found a decrease in the Indian region from past analysis of summer PR of about 47% and mainly attributed to local land-atmosphere interactions. These results are much larger than the ones found here, on average a 3.34% decrease in summer rainfall in the past analysis (1950–2014). Ref. [38] evidenced an even larger reduction in PR of 60% during the post-monsoon season. Ref. [46] also found a larger range of summer PR changes (around 1.5 mm/day) than our study (average of 0.22 mm/day). Ref. [41] attributed a 7.4% decrease in early summer precipitation to irrigation in the Indian subcontinent, which was the closest to our mean values (3.34%). The differences in the magnitude of PR changes between the studies confirmed how difficult it is to properly resolve processes involved in controlling rainfall in monsoon regions. Nonetheless, the importance of the local land-atmospheric feedback in shaping rainfall in these regions was supported by our study and previous ones. Our results did not allow us to conclusively attribute the differences in rainfall changes to potential remote effects via teleconnections or to the use of an interactive ocean. However, they suggest that those aspects did have an impact on the final LULCC-induced effect on precipitation changes.

## 5. Conclusions

This paper aimed to identify the BGP effects of LULCC on the climate in South Asia. The research identified the annual and seasonal mean changes in near-surface temperature and total precipitation over the region by assessing 11 CMIP6-GCMs models. Even though previous studies have already shown how irrigation and other LULCC could modulate climate in the region, the current study highlighted the full range of the BGP effects from LULCC by including requirements that were previously overlooked. The multimodel analyses from CMIP6-GCMs showed that the LULCC promoted robust surface net cooling and surface net drying in particular parts of South Asia at all seasons, and wetting in Northwest Pakistan.

LH, Esurf, and Qa represented the dominant flux and variables to surface cooling by driving changes in TAS. The PR decrease was a result primarily attributed to changes in the local hydrological recycling processes but was also intimately connected to forces beyond the local changes.

Atmospheric feedback contribution to cooling was particularly stronger during winter and spatially dominant during post-monsoon season. The findings from CMIP6-GCMs were comparable in terms of PR and TAS changes between the single model with irrigation representation (CESM2) and the other models without irrigation. This outcome might reveal that, when summing all the BGP interactions, the net effect from other LULCC may be masked by irrigation, atmospheric feedback, or even by the combination of both. Therefore, more models with the representation of irrigation must be included for a definitive assessment of full BGP effects when land management is part of the definition of land-use and land-cover changes.

Our findings overall agreed with previous studies on the direction of change. However, the magnitude of changes may differ from one to another due to various reasons. The most probable of these are, on the one hand, the CMIP6-GCMs' coarse resolution and wet/cold biases, and on the other hand, the fact that previous studies have disregarded the atmospheric feedback contribution, the interaction with the ocean component, the interannual land-use transitions, and even the inclusion of land management into the characterization of LULCC.

Nevertheless, there were still limitations to this study. The coarse nature of GCMs prevented detailed information on specific land processes especially that linked to PRconv. The mechanisms involved in complex topographical areas were not well-resolved in those GCMs, hence further studies are necessary for these regions. A similar experimental setup but with high resolution and systematic inclusion of irrigation would benefit in closing some of the gaps from the CMIP6-GCMs. Knowing that high-resolution simulation runs are usually too costly, an alternative was the use of satellite images to infer potential changes in climate variables by comparing areas with similar background climates and contrasting vegetation. Moreover, the analysis of model responses did not allow us to conclude on the complete causes of the decrease in rainfall, particularly the summer precipitation. Even so, the findings encompassed 11 different models and were calculated over a long temporal scale (65 years), both increasing the robustness of a signal.

LULCC may alleviate warming in South Asia, but these changes could also enhance eminent drying in the region. That could pose further dependency, in some areas, on irrigation and continuous drying. Besides, LULCC affect other biophysical niches, beyond the climate system, which can inflict additional constraints on sustainability. An increase in irrigation, for example, may not only be necessary for the frame of food security policies but may exaggerate water scarcity and water conflicts. A careful and complete examination should be considered when planning and managing land and its potential in adaptation plans for reducing climate-related vulnerabilities in South Asia.

**Supplementary Materials:** The following supporting information can be downloaded at: https://www.mdpi.com/article/10.3390/land12040880/s1, Figure S1: Discrepancies from LUH2; Figure S2: Bare area changes; Figure S3: Intermodel spread of the feedback strength.

**Author Contributions:** Conceptualization, J.F.S., U.S. and S.u.H.; Methodology, J.F.S. and S.u.H.; Software, J.F.S.; Formal Analysis, J.F.S.; Investigation, J.F.S., U.S., S.u.H. and J.B.; Resources, J.F.S., U.S. and S.u.H.; Data Curation, J.F.S.; Writing—Original Draft Preparation, J.F.S.; Writing—Review and Editing, J.F.S., U.S. and S.u.H.; Visualization, J.F.S.; Supervision, U.S. and S.u.H.; Project Administration, J.F.S. All authors have read and agreed to the published version of the manuscript.

**Funding:** This research has been funded by the Deutsche Forschungsgemeinschaft (DFG, German Research Foundation) under Germany's Excellence Strategy—EXC 2037 'CLICCS—Climate, Climatic Change, and Society'—Project Number: 390683824, contribution to the Center for Earth System Research and Sustainability (CEN) of Universität Hamburg.

**Data Availability Statement:** The study analyzed publicly available datasets. The CMIP6 simulation results can be found at the German ESFG portal, dkrz CMIP6 search dataset "https://esgf-data.dkrz.de/search/cmip6-dkrz/ (last accessed 21 November 2022)". The land use forcing dataset (LUH2) can be found at the global ecology laboratory—University of Maryland's website "https://luh.umd.edu/data.shtml (accessed on 22 December 2021)".

**Acknowledgments:** The authors acknowledge the World Climate Research Programme (WCRP), and the Earth System Grid Federation (ESGF) for archiving and enabling online public access to the CMIP6 dataset. The data analysis and all computational work carried out in this study were performed on the high-performance computers HLRE3 (Mistral) and HLRE4 (Levante) supercomputer at the Deutsches Klimarechenzentrum (DKRZ). We also thank Remon Sadikni and the Integrated Climate Data Center (ICDC) for helping with the codes used in the analysis and David Lawrence for clarifying issues pertinent to the CMIP6 and LUMIP model outputs as well as the other modelers involved in the CMIP6 research program.

**Conflicts of Interest:** The authors declare no conflict of interest. The sponsors had no role in the design, execution, interpretation, or writing of the study.

## Appendix A

**Table A1.** CMIP6-GCMs model description.

| No. | CMIP6 Model Name | Land Surface Model | Country | Horizontal Resolution (Lon. by Lat. in Degrees) | Reference | Land Use Representation | | | Vegetation Structure | |
|---|---|---|---|---|---|---|---|---|---|---|
| | | | | | | Irrigation | Cropland | Pasture | LAI | Vegetation Distribution |
| 1 | ACCESS-ESM1-5 (ACCESS) | CABLE 2.4 | Australia | $1.9° \times 1.2°$ | [66] | no | yes | no | prognostic | prescribed |
| 2 | BCC-CSM2-MR (BCC) | BCC-AVIM 2.0 | China | $1.1° \times 1.1°$ | [67] | no | Not included | Not included | prescribed | prescribed |
| 3 | CanESM5 (CanESM5) | CLASS 3.6-CTEM 1.2; Physics—CLASS 3.6 Biogeochemistry—CTEM 1.2 | Canada | $2.8° \times 2.8°$ | [68] | (Not output) | Not included | Not included | Not included | prescribed |
| 4 | CESM2 | CLM 5.0 | USA | $1.3° \times 0.9°$ | [69] | yes | yes | no | prognostic | prescribed |
| 5 | CMCC-ESM2 (CMCC) | CLM 4.5 | Italy | $1.3° \times 0.9°$ | Not available | no | no | no | prognostic | prescribed |
| 6 | CNRM-ESM2-1 (CNRM) | ISBA-CTRIP | France | $1.4° \times 1.4°$ | [70] | no | yes | no | prognostic | prescribed |
| 7 | GFDL-ESM4 (GFDL) | GFDL-LM 4.1 | USA | $1.3° \times 1°$ | [71] | no | yes | yes | prognostic | prescribed |
| 8 | IPSL-CM6A-LR (IPSL) | ORCHIDEE v2.0 | France | $2.5° \times 1.3°$ | [72] | no | yes | no | prognostic | prescribed |
| 9 | MIROC-ES2L (MIROC) | MATSIRO 6.0 +VISIT-e v1 | Japan | $2.8° \times 2.8°$ | [73,74] | no | yes | yes | prognostic | prescribed |
| 10 | MPI-ESM1-2-LR (MPI) | JSBACH 3.2 | Germany | $1.9° \times 1.9°$ | [75] | no | yes | yes | prognostic | Simulated |
| 11 | UKESM1-0-LL (UKESM) | JULES-ES-1.0 | UK | $1.9° \times 1.3°$ | [76] | (Not output) | yes | yes | Not included | simulated |
| 12 | LUH2 (Land Use forcing dataset) | GLM 2 (Global Land-Use Model) | - | $0.25° \times 0.25°$ | [77] | yes | yes | yes | | - |

**Table A2.** List of fluxes and other variables from CMIP6-GCMs.

| Variable Name | CMIP6 Variable Code | Unit | Description |
|---|---|---|---|
| Near-surface air temperature | tas (TAS) | Kelvin | Near-surface air temperature usually at 2 m. |
| Precipitation flux | pr (PR) | $Kg\ m^2\text{-}s^{-1}$ | Precipitation flux including both liquid and solid phases. |
| Convective precipitation flux | prc (PRconv) | $Kg\ m^2\text{-}s^{-1}$ | Convective precipitation at surface. It includes both liquid and solid phases. |
| Soil Moisture upper 10 cm | mrsos (SM) | $kg\ m^2$ | The mass of water in all phases in a thin surface layer integrated over the uppermost 10 cm of the soil layer. |

**Table A2.** *Cont.*

| Variable Name | CMIP6 Variable Code | Unit | Description |
|---|---|---|---|
| Total Soil Moisture | mrso (SM10) | kg m$^2$ | The mass per unit of area (summed over all soil layers) of water in all phases. |
| Water Evaporation flux from canopy | evspsblveg (Esoil) | Kg m$^2$-s$^{-1}$ | The canopy evaporation and sublimation (if present in the model). It may include dew formation as a negative flux. |
| Water evapotranspiration flux | evspsbl (Esurf) | Kg m$^2$-s$^{-1}$ | Evapotranspiration at the surface. The flux of water into the atmosphere due to conversion of both liquid and solid phases to vapor. |
| Water evaporation flux from soil | evspsblveg (Esoil) | Kg m$^2$-s$^{-1}$ | Water evaporation flux from soil including sublimation. |
| Surface Upward latent heat flux | hfls (LH) | W m$^2$ | Surface upward latent heat flux. Surface means the lower boundary of the atmosphere and "upward" indicates a vector component that is positive when directed upward. The surface latent heat flux is the exchange of heat between the surface and the air on account of evaporation. |
| Surface upward sensible heat flux | hfss (SH) | W m$^2$ | The surface sensible heat flux, also called turbulent heat flux, is the exchange of heat between the surface and the air by the motion of air. |
| Atmosphere mass content of water vapor | prw (WV) | kg m$^2$ | Water vapor path vertically integrated through the atmospheric column. |
| Leaf area index | lai (LAI) | Unitless | A ratio obtained by dividing the total upper leaf surface area of vegetation by the horizontal surface area of the land on which it grows. |
| Surface downwelling shortwave flux in air | rsds | W m$^2$ | Surface solar irradiance for UV calculations. |
| Surface upwelling shortwave flux | rsus | W m$^2$ | Shortwave radiation from below. |
| Surface downwelling longwave flux in air | rlds | W m$^2$ | Longwave radiation from above. |
| Surface upwelling longwave flux in air | rlus | W m$^2$ | Longwave radiation from below. |
| Cloud area fraction | clt (Cc) | % | Total cloud area fraction for the whole atmospheric column, as seen from the surface or the top of the atmosphere. It includes both large-scale and convective clouds. |

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
