# Peer review of "Biogeophysical Effects of Land-Use and Land-Cover Changes in South Asia: An Analysis of CMIP6 Models"

_land, doi:10.3390/land12040880_

Round 1

Reviewer 1 Report

Dear Authors, I have reviewed the PDF of your manuscript and provided feedback and comments. However, I have identified several areas that require revision for your manuscript to meet the expected standards. Please take my comments into consideration during your revision process. Regards.

Author Response

Thank you for your feedback and constructive comments on how to improve the manuscript. All of the suggestions/comments were taken into consideration, especially the one on the statistical analyses of the relationships.

- We added coordinates in figures 2 and 5 and also increased the resolution of the images. We have not inserted a scale bar nor a north arrow, since the grid cell size (indicated in the methods section) already specifies the scale of the map. Please, be aware that the produced maps derive from global climate model outputs, which have a slightly different format than typical geographical maps.

- The methods section was re-structured as suggested, and a flowchart was added to help the reader better following the steps.

- The whole text was revised for English writing and clarity.

- We add a statistical analysis of the relationships between the TAS/PR dependent variables related to land surface variables/fluxes (independent variables) as described in the methods section.

- Abstract was re-written

Reviewer 2 Report

In this paper, the authors aimed to identify the biogeophysical effects of Land Use, Land Cover and Land Management Changes on the climate in South Asia. The study has identified the annual and seasonal mean changes in near-surface temperature and total precipitation over the area considered by assessing 11 CMIP6-GCMs models. As a whole, the study is interesting and carries forward a clear message for the journal readers interested in climate change. The introduction is very good; the authors demonstrate a deep knowledge of the published literature and highlight the importance and background to carry out this investigation. The methods used are relevant and the results are nicely presented and given in a logical order.

I have only some concerns about the abstract, which is not easily readable since it contains too many acronyms and does not explicitly describe the aim and the main findings. I recommend a general review of the abstract and then the manuscript could be published.

Author Response

Thank you for your comments and a great/positive summary of our manuscript. About the abstract, it indeed sounds confusing and we improved it, also got rid of some of the abbreviations since they are introduced in the main text anyway.

We also improve the overall paper by inserting a statistical analysis of relationships between TAS/PR dependent variables related to land surface variables/fluxes (independent variables) as described in the methods section. Accordingly, some images and content of the result section in the previous version were modified.

Round 2

Reviewer 1 Report

Dear Authors,

Thank you for your efforts to revise your manuscript. However, I do have a few comments that the authors should take into account when preparing the final draft of their manuscript.

Regards.
